

# Improving soil aquifer treatment efficiency using air injection into the subsurface

Ido Arad[1], Aviya Ziner[1] Shany Ben Moshe[1], Noam Weisbrod[2], and Alex Furman[1]

[1]Civil and Environmental Engineering, Technion – Israel Institute of Technology, Haifa 32000, Israel
[2]The Zuckerberg Institute for Water Research, Jacob Blaustein Institutes for Desert Research,
Ben-Gurion University of the Negev, Midreshet Ben-Gurion, 8499000, Israel

*Correspondence to:* Ido Arad (idoarad1@gmail.com)

**Abstract.** Soil aquifer treatment (SAT) is an effective and sustainable technology for wastewater or stormwater treatment, storage and reuse. During SAT, the vadose zone acts as a pseudo reactor in which physical and biochemical processes are utilized to improve the infiltrated water quality. Dissolved oxygen (DO) is necessary for aerobic microbial oxidation of carbon and nitrogen species in the effluent. Therefore, to enhance aeration, SAT is generally operated in flooding and drying cycles. While long drying periods (DPs) lead to better oxidizing conditions and improve water quality, they reduce recharge volumes. As the population grows, the quantity of effluent directed to SAT sites increases and increasing recharge volumes become a concern and often a limiting factor for SAT usage.

In this study, direct subsurface air injection SAT (Air-SAT) was tested as an alternative to long DPs operation. Six long column experiments were conducted, aiming to examine the effect of air injection on the soil's water content, oxidation-reduction potential (ORP), DO concentrations, infiltrated amounts and ultimate outflow quality. In addition to basic parameters such as dissolved organic-C (DOC) and N species, the effluent quality analysis also included an examination of three emerging water contaminants – Ibuprofen, Carbamazepine and 1H-benzotriazole. Pulsed air injection experiments were conducted during continuous flooding at different operation modes (i.e., air pulse durations, frequencies and airflow rates).

Our results show that the Air-SAT operation doubled the infiltration time (i.e., the infiltration was continuous with no off-time) and allowed up to 46% higher infiltration rate in some cases. As a result, the infiltrated volumes in the Air-SAT modes were 47-203% higher than the conventional flooding-drying operation (FDO). Longer air pulse duration (60 vs. 8 min) and higher airflow rate (~2 vs. ~1 SLPM) led to a higher infiltration rate, while a high pulse frequency ($4.5^{-1}$ $h^{-1}$) led to a lower infiltration rate compared to low-frequency operation ($24^{-1}$ $h^{-1}$).

The air injection also allowed good recovery of the ORP and DO levels in the soil, especially in the high-frequency Air-SAT experiments, where steady aerobic conditions were maintained during most of the flooding. Consequently, the mean DOC,

total Kjeldahl N (TKN), and Ibuprofen removals in these experiments were higher than in FDO by up to 9, 40, and 65%, respectively. However, high-frequency Air-SAT during continuous flooding also led to significant deterioration in the infiltration rate, probably due to enhanced biological clogging. Hence, it may be more feasible and beneficial to combine it with the conventional FDO, allowing a steady infiltration rate and increased recharge volumes, while sustaining high effluent quality. While those results still need to be verified at full scale, they open the possibility of using air injection to minimize the DPs length and alleviate the pressure over existing SAT sites.

## 1 Introduction

With the growing global population and an increased need for water, wastewater (WW) reuse has become essential worldwide, especially in arid and semi-arid regions (Montwedi et al., 2021; Qureshi, 2020; Steduto et al., 2012). Soil aquifer treatment (SAT), a subfield of managed aquifer recharge (MAR), is a nature-based, robust and cost-effective system for WW or stormwater treatment, storage and reuse (Sharma and Kennedy, 2017). SAT systems usually involve a cluster of infiltration ponds surrounded by production and monitoring wells (Idelovitch and Michail, 1984; Sharma and Kennedy, 2017). The feed water is intermittently infiltrated through a deep vadose zone into the aquifer. During the infiltration, physical and biochemical processes (e.g., filtration of suspended solids, adsorption to the soil components and biodegradation) occur, and water quality is greatly improved (Gharoon and Pagilla, 2021; Goren et al., 2014). The reclaimed effluent is stored in the aquifer and can be used for many applications. For example, using an SAT system, The Dan region WW treatment plant (SHAFDAN) in Israel produces more than 160 million $m^3$ $year^{-1}$ of reclaimed effluent used for unrestricted crop irrigation (Aharoni et al., 2020).

SAT takes advantage of various microbial-mediated processes that take place during the infiltration. Although physical and chemical mechanisms (e.g., adsorption) can, in principle, remove organic pollutants in SAT, the primary mechanism for removing dissolved organic matter (DOM) is biodegradation (Quanrud et al., 1996; Rauch and Drewes, 2005). This is because the organic load is too high for efficient removal only by physical and chemical mechanisms. Therefore, increasing the SAT biodegradation efficiency is of uppermost importance (Brooks et al., 2020). The DOM can be oxidized aerobically, using $O_2$ as the terminal electron acceptor or by other terminal electron acceptors such as $NO_3^-$ (i.e., denitrification; Goren et al., 2014). However, the DOM removal in SAT is more efficient under aerobic conditions (Ben Moshe et al., 2021)

The effluent that enters SAT system typically contains several forms of N, such as organic-N, $NH_4^+$, $NO_2^-$ and $NO_3^-$ (Ickeson-Tal et al., 2003; Bouwer, 1991). Organic-N ammonification occurs in both aerobic and anaerobic conditions (Stefanakis et al., 2014), but its rate is higher under aerobic conditions since ammonifying bacteria populations thrive under high dissolved oxygen (DO) levels (Ruan et al., 2009). Aerobic conditions in SAT are also essential for nitrification - the oxidation of $NH_4^+$ to $NO_2^-$ and the subsequent oxidation of $NO_2^-$ to $NO_3^-$ (Mienis and Arye, 2018). N can be removed from the effluent by filtration of suspended solids, along with the adsorption of N compounds to the soil minerals and organic matter (van Raaphorst and

Malschaert, 1996; Idelovitch et al., 2003). In addition, N can also be removed by biochemical processes – microorganisms in
the soil assimilate $NH_4^+$ and $NO_3^-$ into their tissues, while denitrifying microbes can reduce $NO_3^-$, ultimately producing $N_2$ gas
(Jetten, 2008).

In conclusion, $O_2$ is necessary for various biogeochemical processes that enhance the effluent quality in SAT. Therefore, in
order to maintain an efficient process, SAT is generally operated in cycles of flooding and drying. The drying periods (DPs)
were designated mainly to recover the gaseous oxygen, and through it the DO levels in the soil profile by allowing air to enter
the soil (Icekson-Tal et al., 2003). Ben Moshe et al. (2020) showed that longer DPs led to better aeration in the deeper parts of
the soil profile and, consequently, to lower outflow concentrations of dissolved organic-C (DOC), dissolved organic-N (DON)
and $NH_4^+$. Sallwey et al. (2020) found a similar trend – the removal of six examined contaminants of emerging concern (CECs)
increased significantly when the DPs became longer. However, long DPs allow a shorter time for infiltration and smaller
recharge volumes. As the population grows, the quantity of effluent directed to SAT sites increases while land resources
become precious. In such conditions, long DPs become less feasible.

This study explored the ability to actively inject air into the subsurface to enhance DO availability in SAT (Air-SAT). This
approach, that to the best of our knowledge was not tested before in the context of SAT, may be an alternative to long DPs
operation, allowing higher reclaimed effluent quantities without compromising water quality. We examined, at the laboratory
scale, the effect of the air injection on the infiltrated volumes, soil biogeochemical state, and ultimate outflow quality. In
addition to macro pollutants (DOM and N species), we also explored the removal of three well-known CECs that are commonly
detected in effluent: the anti-inflammatory medication Ibuprofen (IBP), the anticonvulsant medication Carbamazepine (CBZ)
and the corrosive inhibitor 1H-benzotriazole (BTR).

## 2 Materials and methods

### 2.1 The experimental System

A 200 cm high polycarbonate column consisting of 20 modules was designed and constructed. The dimensions of each module
were 20X10X10 cm. The top three and base modules were kept empty for flooding and drainage, respectively. The 16
remaining modules were packed with sandy soil from the upper meter of the SHAFDAN SAT site in Israel (Fig. 1a). The soil
texture (96.4% sand, 2.1% silt, 1.5% clay) was determined using the hydrometer method (Gee and Or, 2002), and the initial
total organic C (TOC) content (0.87%) was determined using the loss on ignition method (Dean, 1974).

Synthetic effluent was prepared and stored in a stirred plastic container with volume scale marks. It was introduced into the
column using a peristaltic pump. The top of the column had an overflow outlet, which enabled a maximal head of ~23 cm.
Compressed air was supplied to the system by an air pump and injected at a depth of 85 cm (below the soil surface level) using



a buried air stone. The volumetric airflow rate was controlled using a digital flow controller, which also measured the air temperature (MC-10SLPM, Alicat Scientific, Tucson, AZ, USA). The injection pressure was measured by a digital pressure controller (serving as a pressure meter), Alicat model PC-30PSIG (Fig. 1a). Air injection rate, temperature, and pressure were recorded at 0.5-minute intervals. Using these data and assuming air to be an ideal gas, the standard volumetric flow rate (i.e., the volumetric flow rate at standard conditions of T = 273.15 K and P = 1 bar) was calculated (Eq. S1).

Four measurement ports were located at depths of 25, 65, 105, and 145 cm below soil surface (Fig. 1a). Each port was equipped with a volumetric water content (VWC) sensor (TDR-315H, Acclima, Meridian, ID, USA), an oxidation-reduction potential (ORP) sensor (Art. No. 461, ecoTech, Bonn, Germany), a rhizon for pore-water sampling (MOM 10 cm, Rhizosphere, Wageningen, Netherlands), and a luminescent dissolved oxygen (LDO) probe (LDO10101, Hach-Lange GmbH, Düsseldorf, Germany). An Ag/AgCl ORP reference electrode filled with 3 mol L$^{-1}$ KCl electrolyte (ecoTech model 4622) was located only at the deepest port (Fig. 1b). In addition, a pre-calibrated pressure sensor (MPX2010DP, NXP Semiconductors, Eindhoven, Netherlands) was used to measure the surface pressure head. The sensors' data were collected every 1-minute using a CR1000 datalogger (Campbell Scientific, Logan, UT, USA), except for the oxygen data, which were recorded using two HQ40d meters (Hach-Lange GmbH).





**Fig. 1. (a)** The experimental system with location of the measuring ports and the air injection port. The ports are referenced in the text as their depth below ground level (cm). The effluent inflow system is highlighted in blue (thick lines): w1 – effluent container, w2 – peristaltic pump, and w3 – overflow hose. The air system is highlighted in red (thin lines): a1 – air pump, a2 – digital flow controller, a3 – digital pressure meter, and a85 – injection port. **(b)** Installation of the devices at each measuring port: b1 - VWC sensor (TDR), b2 – ORP sensor, b3 – rhizon, b4 –reference electrode, and b5 – LDO sensor.

## 2.2 Column experiments

Six 72-hour column experiments were conducted. All experiments started when there was no pressure head above the unsaturated soil surface and after long drainage of at least nine days. Effluent supply was at higher rate than the infiltration, and at the beginning of each flooding period (FP), the surface head (SH) increased for ~1 hour until it reached a maximal value of ~23 cm and remained constant for the rest of the FP.

The six experiments are divided here into three main and three secondary experiments. The first main experiment included a 24-hour FP followed by a 24-hour DP and another 24 hours of flooding. It was operated without any active air injection (Table 1). The term FP refers to the duration of time during which effluent was pumped to the top of the column. The DP started as the pump was turned off and ended at the beginning of the second FP. This experiment, which was conducted twice, is noted here as FDO (flooding-drying operation), and it represents the conventional intermittent operation used in SAT sites.





The two other main experiments were designed to examine the effect of air injection, at different operation modes (i.e., pulse durations and frequencies), on the biogeochemical efficiency of the SAT system. Both included one continuous 72-hour FP during which compressed air was injected at 85 cm depth in pulses, at a rate of ~2 SLPM. They are noted here as AI-LF$_1$ and AI-HF$_1$, where the abbreviation AI stands for air injection (Table 1). LF and HF denote the pulse frequency (low and high, respectively). During AI-LF$_1$, the air was injected into the subsurface for 60 minutes every 24 hours of flooding. In contrast, during AI-HF$_1$, the air was injected for only 8 minutes, but the pulse frequency was significantly higher – one pulse every 4.5 hours of flooding. These specific pulse durations and frequencies were designed to achieve equal volumes of injected air in each of the two experiments (Table 1).

**Table 1.** The operational parameters of the three main and three secondary experiments. SLPM and SL stand for standard liters per minute and standard liters, respectively (i.e., LPM and L at standard conditions of T = 273.15 K and P = 1 bar)

|  | Experiment | FP / DP (h) | Average airflow rate (SLPM) | Pulse duration (min) | Pulse frequency (h$^{-1}$) | Total injected air (SL) |
|---|---|---|---|---|---|---|
|  | FDO | 24:24 | - | - | - | 0 |
| Main | AI-LF$_1$ | Only flooding | 2.037 | 60 | 24$^{-1}$ | 244 |
|  | AI-HF$_1$ | Only flooding | 2.001 | 8 | 4.5$^{-1}$ | 240 |
|  | AI-HF$_2$ | Only flooding | 0.991 | 16 | 4.5$^{-1}$ | 238 |
| Secondary | AI-HF$_3$ | Only flooding | 0.980 | 8 | 4.5$^{-1}$ | 118 |
|  | AI-LF$_2$ | Only flooding | 2.010 | 8 | 24$^{-1}$ | 32 |

The secondary experiments AI-HF$_2$ and AI-HF$_3$ aimed to examine the impact of the airflow rate. The airflow rate in both of them was approximately half (~1 SLPM) than in the main experiment AI-HF$_1$ (~2 SLPM), while the pulse frequency was similar - 4.5$^{-1}$ h$^{-1}$. In AI-HF$_3$, the pulse duration was identical to AI-HF$_1$ (8 min), which effectively means that approximately half the air volume was injected. In AI-HF$_2$, the duration was doubled (16 min) in order to get a similar overall volume of injected air (Table 1).

The secondary experiment AI-LF$_2$ was designed to connect the main experiments AI-LF$_1$ and AI-HF$_1$. This experiment included short 8-min pulses (similarly to AI-HF$_1$) injected at low frequency (every 24 hours, like in AI-LF$_1$) and an airflow rate of ~2 SLPM. Note that in comparison to the three other air-injection experiments, the volumes of air injected in AI-HF$_3$ and AI-LF$_2$ were smaller (118 SL and 32 SL, respectively; Table 1).





Effluent samples along the profile were collected twice a day at approximately identical times for each experiment ($t \sim 4.3$,
23.7, 28.4, 47.7, 52.4, 71.6 h, where $t = 0$ is the beginning of each experiment). In the FDO experiment, samples were not
taken at $t \sim 28.4$, 47.7 as the soil was too dry. At the same times, the inflow effluent was also sampled to confirm that no
significant changes in its composition occurred during the experiments. The infiltrated volumes were measured manually by
reading the container's scale marks, and the mean infiltration flux (cm h$^{-1}$) was calculated as the infiltrated volume over a given
time interval (cm$^3$ h$^{-1}$) divided by the cross-sectional area of the chamber ($A = 195.03$ cm$^2$). After each experiment, the column
was flushed with tap water for 12 hours to restrict clogging and was left to dry for several days until mean ORP levels recovered
above 185 mV (i.e., aerobic conditions prevailed along the column). The mean ORP was calculated as the arithmetic mean of
the ORP measured at the four ports along the column.

### 2.3 Synthetic effluent composition and preparation

The synthetic effluent included mainly $NH_4^+$, Glucose and Urea dissolved in tap water. Its composition was designed to include
a moderate load of DOC as well as organic and inorganic N species around the concentrations found in the SHAFDAN SAT
sites (Aharoni et al., 2020). Glucose was the primary C source (accounted for ~99% of the DOC). $NH_4^+$, total Kjeldahl N
(TKN), $NO_3^-$, total N (TN), and DOC concentrations for the synthetic effluent, analyzed throughout the main experiments, are
presented in Table 2. In addition, the synthetic effluent also included the emerging contaminants IBP, CBZ and BTR at low
levels, around the concentrations found in effluent collected from a municipal WW treatment plant in Germany (Table 2;
Sallwey et al., 2020). Characterization of the synthetic effluent for each experiment separately (including the secondary
experiments) is available in the supplement (Table S3, S4). To prepare the effluent, the following chemicals were used:
Ammonium Chloride ($NH_4Cl$, >99.5%, Spectrum), D-(+)-Glucose monohydrate ($C_6H_{12}O_6 \cdot H_2O$, >97.5%, Sigma-Aldrich),
Urea ($CH_4N_2O$, >99%, Sigma-Aldrich), Ibuprofen ($C_{13}H_{18}O_2$, >98%, Sigma-Aldrich), Carbamazepine ($C_{15}H_{12}N_2O$, >98%,
Sigma-Aldrich) and 1H-Benzotriazole ($C_6H_5N_3$, >99%, Acros Organics).

**Table 2.** Inflow composition of the synthetic effluent in the three main experiments – mean ± SD

| [$NH_4^+$] (mg-N L$^{-1}$) | [TKN] (mg L$^{-1}$) | [$NO_3^-$] (mg-N L$^{-1}$) | [TN] (mg L$^{-1}$) |
|---|---|---|---|
| 2.62 ± 0.98 | 8.74 ± 0.56 | 0.85 ± 0.66 | 9.60 ± 0.93 |
| [DOC] (mg L$^{-1}$) | [IBP] (μg L$^{-1}$) | [CBZ] (μg L$^{-1}$) | [BTR] (μg L$^{-1}$) |
| 41.20 ± 1.36 | 1.13 ± 0.29 | 1.15 ± 0.31 | 8.79 ± 1.38 |



## 2.4 Chemical analysis

Previously to the chemical analysis, all effluent samples were passed through a 0.22 µm filter. $NH_4^+$ was measured by a colorimetric method (Willis et al., 1996) using a Genesys 150 Spectrophotometer (Thermo Scientific, Waltham, MA, USA). $NO_2^-$ and $NO_3^-$ were measured using an ion chromatograph (881 Compact IC pro, Methrom AG, Herisau, Switzerland). DOC and TN were determined using a total organic C analyzer (TOC-VCPH) equipped with a TN module unit (TNM-1) (Shimadzu, Kyoto, Japan) after the samples were acidified with HCl to achieve pH = 2-3. TKN concentrations were calculated by the difference between the concentrations of TN and the oxidized N ($NO_2^-$ and $NO_3^-$):

$$(1) \ [TKN] = [TN] - ([NO_2^-] + [NO_3^-])$$

where $[TKN]$, $[TN]$, $[NO_2^-]$ and $[NO_3^-]$ are all in mg-N L$^{-1}$.

IBP, CBZ and BTR were analyzed using a liquid chromatograph (Agilent 1110, Agilent Technologies, Santa Clara, CA, USA) coupled to a triple-quadrupole mass spectrometer (API 3200, Applied Biosystems Sciex Instruments, Waltham, MA, USA) equipped with electrospray ionization (ESI). A RP-18 end-capped column with 5-µm particle size (Purospher® Star, Merck, Darmstadt, Germany) was used at 30°C for separation. The eluent flow rate was 0.4 mL min$^{-1}$ and the injection volumes were 20 µL. A detailed description of the LC-MS/MS analytical procedures appears in the supplement.

## 3 Results and discussion

### 3.1 Effluent infiltration

Figure 2 compares the VWC at a depth of 25 cm below soil surface ($\theta_{25}$) and the mean infiltration flux between the experiments FDO (intermittent flooding-drying operation), AI-LF$_1$ (60-minute long, ~2 SLPM air pulse, every 24 hours of continuous effluent infiltration) and AI-HF$_1$ (8-minute long, ~2 SLPM pulse, every 4.5 hours of continuous infiltration). During the first 24 hours of FDO and AI-LF$_1$, when there was no active air injection into the subsurface, the infiltration flux was similar between those two experiments. The first VWC increase (i.e., the arrival of the wetting front) was observed after 12 and 18 minutes, while another increase can be noticed after 14 and 20 hours in FDO and AI-LF$_1$ (less pronounced), respectively. At the end of the first 24 hours, $\theta_{25}$ reached apparent steady-state conditions, with water contents of 35.4% in FDO and 27.5% in AI-LF$_1$ (Fig. 2, panels a and b). The difference between the two treatments is possibly due to a more significant air entrapment during AI-LF$_1$. The impact of air entrapment on the soil's water content during water infiltration was demonstrated by Mizrahi et al. (2016).





After 24 hours of flooding, air was injected for 60 minutes in AI-LF$_1$, and $\theta_{25}$ slightly decreased (Fig. 2b), while a more significant reduction in the VWC was observed at depths of 65, 105 and 145 cm (Fig. S1b, S2b and S3b, respectively). An

immediate reduction in the VWC along the profile was also detected during the second pulse (48 h; Fig. 2b, S1b, S2b, S3b). This phenomenon (i.e., a reduction in the VWC as a result of the injection of pressurized air into sandy soil) was demonstrated in numerous studies (e.g., Dror et al., 2004; Ben-Noah et al., 2021; Zang and Li, 2021) and can be explained by that the air pushed the effluent away from its flow pathways. As the unsaturated hydraulic conductivity is an increasing function of the VWC (van Genuchten, 1980), a reduction in the infiltration flux during the air pulses, as a result of the creation of low-

conductivity zones, could be expected. Indeed, during the first pulse, the infiltration flux was decreased by half – from a mean value of 20.51 cm h$^{-1}$ in the first 24 hours to 10.25 cm h$^{-1}$ during the pulse. During the second pulse (48 h), this reduction was even more drastic (-65%; Fig. 2b).

Between the two air pulses in AI-LF$_1$, $\theta_{25}$ increased from 26.3% to 45.8%. This value (i.e., $\theta_{25}$ = 45.8%) is 18.3% higher than

the steady-state VWC reached at this depth before the first pulse ($\theta_{25}$ = 27.5%). A similar phenomenon, of a sharp increase in water content, was observed following the second pulse (Fig. 2b). We suggest that the air injection formed new preferential pathways (PFP) for both effluent and air, allowing on one hand the release of entrapped air and on the other creating wider pores. The air pulses possibly helped break surface and sub-surface crusts which are often formed and cause clogging during the operation of SAT systems. Such crusts can be formed by chemical, physical or biological factors, e.g., chemical

precipitation of compounds found in the effluent, accumulation of suspended solids and growth of biofilm (Barry et al., 2017; Pavelic et al., 2011; Thuy et al., 2022)

This evidence (i.e., increased VWC following the air pulse) was observed at 60 ($\theta_{25}$; Fig. 2b) and 20 cm ($\theta_{65}$; Fig. S1b) above the air source but was not observed at 20 ($\theta_{105}$; Fig. S2b) and 60 cm ($\theta_{145}$; Fig. S3b) below it, probably because the airflow

pattern was mostly upward due to the buoyancy force. The decrease in $\theta_{105}$ and $\theta_{145}$ during the air injection (Fig. S2b and S3b, respectively) is likely due to the reduced flux above, but its timing indicates also direct air movement downwards.

As expected, in AI-LF$_1$, the increased VWC following the air injection allowed an increased infiltration rate: the flux after the first and second pulse (28.98, 28.31 cm h$^{-1}$, respectively) was 38-41% higher than before the first pulse (20.51 cm h$^{-1}$; Fig. 2b).

Meanwhile in FDO, the infiltration flux in the second FP, (19.44 cm h$^{-1}$, Fig. 2a) was 4% lower compared to the first FP (20.30 cm h$^{-1}$) and 31% lower compared to the last 23 hours of AI-LF$_1$ (Fig. 2b).

In AI-HF$_1$, as in AI-LF$_1$, each air pulse led to an immediate but temporary decrease in the VWC along the profile (Fig. 2c, S1c, S2c, S3c). The infiltration flux, which was higher than the flux in AI-LF$_1$ and FDO during the first 24 hours (23.71 cm h$^{-1}$),

decreased over time. In the last 24 hours, it reached a minimum value of 14.31 cm h$^{-1}$ – 26% lower compared to the last 24 hours of FDO and 49% lower compared to the last 23 hours of AI-LF$_1$ (Fig. 2c). This deterioration can be explained by the





gradual formation of low conductivity clogging layers. The overall trend of decline in the maximal values of $\theta_{25}$ during the last 24 hours of AI-HF$_1$ (Fig. 2c) supports this hypothesis. This trend was not observed during the first 48 hours, although there was a decline in the infiltration flux (Fig. 2c), suggesting that at the beginning of AI-HF$_1$, the clogging processes mainly

occurred at shallower depths. Apparently, the short pulses were less efficient in breaking crusts than the longer ones, as the water content reached values around 40% in this case, compared to around 45% in AI-LF$_1$.

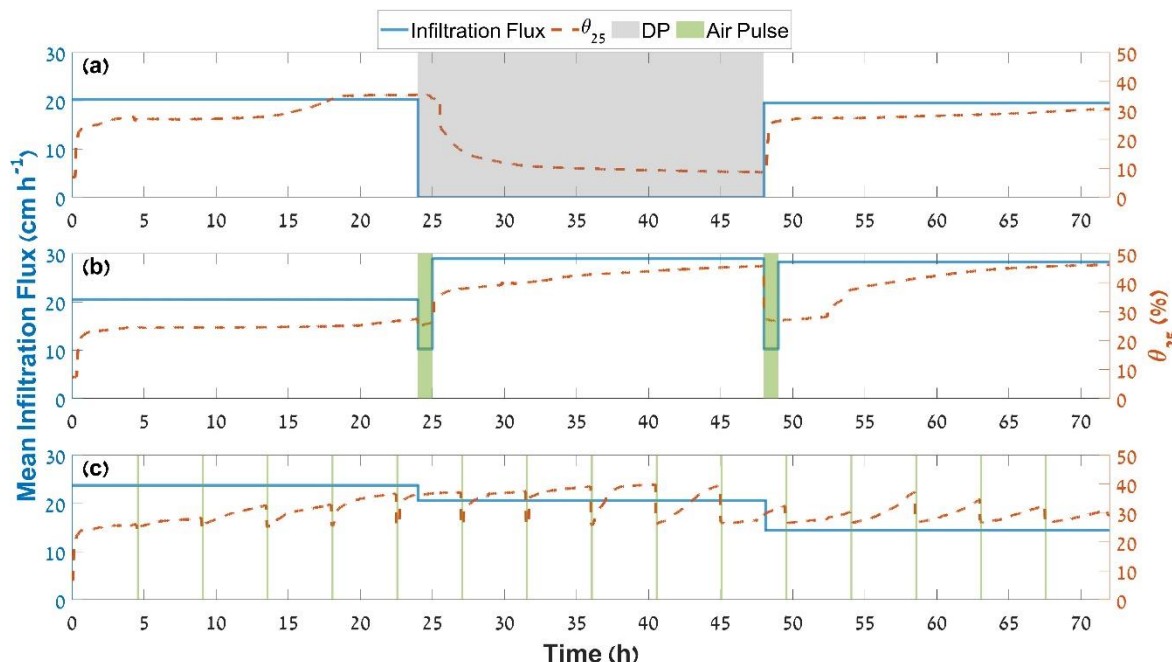

**Fig. 2.** VWC at a depth of 25 cm below soil surface ($\theta_{25}$) and the mean infiltration flux during FDO, AI-LF$_1$ and AI-HF$_1$ (panels **a**, **b** and **c**, respectively). Gray areas symbolize DP, while green areas symbolize the air pulses.

Air-SAT operation allowed much higher recharge volumes compared to the traditional FDO. For fairness of comparison (i.e., not including the early stages that are more affected by the initial conditions), we use only the last 48 hours for comparison between treatments. While in FDO only 86 L were infiltrated, in AI-LF$_1$ and AI-HF$_1$ the amounts were significantly higher (261 and 164 L, respectively; Fig.3). The main reason for this gap is the flooding time – while in AI-HF$_1$ and AI-LF$_1$ the infiltration could be continuous thanks to the air injection, in FDO 24 hours were hydraulically wasted for drying. Isolating

only FDO and AI-HF$_1$, the higher time dedicated to flooding is actually the only reason for this gap, as the mean infiltration flux during the last 48 hours of AI-HF$_1$ (17.52 cm h$^{-1}$) was lower than the flux during the second FP of FDO (19.44 cm h$^{-1}$; Fig. 2a). In AI-LF$_1$, in addition to the doubled flooding time, the significantly higher infiltration rate compared to FDO (Fig. 2) is another reason for the three times higher infiltrated volume (Fig. 3).



In fact, due to the high infiltration rate observed in AI-LF$_1$, the infiltrated volume in this experiment was 59% higher than in AI-HF$_1$ (Fig. 3), although their flooding and aeration times were identical. This leads us to the conclusion that the difference in the infiltration rate is related to the pulse frequency and duration. The high-frequency operation mode (AI-HF$_1$) induced steadier aerobic conditions (Fig. 4), which are more suitable for fast biofilm growth (Lappin-Scott and Bass, 2001; Naz et al., 2013). As a result, the biological clogging might be more significant in this experiment, causing the infiltration rate
deterioration. Note that this deterioration, apparently caused by clogging, also occurred in the less-oxidizing experiments FDO and AI-LF$_1$, but less significantly (Fig. 2). In addition to the increased clogging effect, the high-frequency injection also seemed to physically hinder the soil from reaching water saturation conditions (Fig. 2c), resulting in a reduced infiltration flux.

        An additional possible explanation for the higher infiltration rate in AI-LF$_1$ is the difference in the pulse duration. We
hypothesize that the sixty minutes of active air injection in AI-LF$_1$ might have been more efficient in forming new PFP and breaking surface and subsurface crusts compared to the short eight-minute pulse performed in AI-HF$_1$. The secondary experiment AI-LF$_2$, which shared pulse frequency with AI-LF$_1$ ($24^{-1}$ h$^{-1}$) and duration with AI-HF$_1$ (8 min; Table 1), can help distinguish between these two mechanisms explaining the higher infiltration rate in AI-LF$_1$ (i.e., reduced clogging and enhanced creation of PFP). The volume infiltrated in this experiment was 216 L – 45 L lower than in AI-LF$_1$ but 52 L higher
than in AI-HF$_1$ (Fig. 3). Hence, we conclude that both mechanisms occurred simultaneously, and in our system, long pulses (60 min) lead to a higher infiltration rate than short pulses (8 min), while injecting air at a high-frequency ($4.5^{-1}$ h$^{-1}$) leads to a lower rate than low-frequency injection ($24^{-1}$ h$^{-1}$).

        In addition to the pulse duration and frequency, the airflow rate also influenced the recharge volume. In AI-HF$_1$, the infiltrated
volume was 30% higher (Fig. 3) than in AI-HF$_3$, which was characterized by the same pulse duration (8 min) and frequency ($4.5^{-1}$ h$^{-1}$), but approximately half the airflow rate – 0.980 SLPM (Table 1). These results suggest that like a longer pulse duration, a higher airflow rate is more efficient in forming PFP and opening local clogging, which cause an increased infiltration rate. PFP formation, however, may result in lower effluent quality due to a shorter detention time and less interaction with the degrading microorganisms. Our results hint that this undesired phenomenon truly occurred during AI-LF$_1$
(Fig. 5,6).

        The relatively small infiltrated volume observed in AI-HF$_3$ is mainly due to a gradual decrease in the infiltration rate - from 22.01 cm h$^{-1}$ during the first 24 hours to 15.38 cm h$^{-1}$ during the second and 11.54 cm h$^{-1}$ in the last 24 hours (Table S5). In fact, our results show that high-frequency Air-SAT (represented here by AI-HF$_1$ and AI-HF$_3$) leads to significant deterioration
in the infiltration rate. In the long run, such an operation can significantly decrease the recharge capacity of SAT sites. Therefore, we do not recommend using it without DPs, which, besides DO recovery, also have a significant role in maintaining a steady infiltration flux during SAT operation. The DPs recover the infiltration rate by allowing the drying, cracking and decomposing of clogging layers (Bouwer, 2002) and enabling periodic tillage of the basin's surface, which breaks the clogging




crusts itself and removes disturbing vegetation (Negev et al., 2020). Apparently, air injection cannot induce such mechanisms

efficiently. Hence, we suggest that the active air injection should be incorporated in the conventional intermittent operation, but likely with much longer wetting times and shorter and less frequent DPs than usually operated. This combination of both active and passive aeration can reduce the required drying length while sustaining a steady infiltration rate.



**Fig. 3.** Infiltrated volumes during the last 48 hours of FDO (average of the two replicates), AI-LF$_1$, AI-HF$_1$, AI-HF$_3$ and AI-LF$_2$. The

volumes infiltrated in the first 24 hours are excluded here in order to minimize the impact of the starting conditions.

## 3.2 effluent quality

Figure 4 compares the mean ORP throughout the column (arithmetic mean of the ORP measured at depths of 25, 65, 105 and 145 cm below soil surface; ORP$_{mean}$) between the experiments FDO, AI-LF$_1$ and AI-HF$_1$. While mean values miss the spatial

variation, for the sake of conciseness we present here only them and provide the full ORP trends in the supplement (Fig. S4).

At the beginning of each of the three experiments, ORP$_{mean}$ sharply increased (Fig. 4) due to the drainage of residual water at 145 cm depth (Fig. S3), followed by air penetration and ORP rise at this depth (Fig. S4d). Afterward, ORP$_{mean}$ generally declined with time (Fig. 4) owing to limited aeration of the column, while O$_2$ and other electron acceptors were consumed by

the soil system's microbial community. This expected decline proceeded in AI-LF$_1$ until the beginning of the first air pulse (t = 24 h; Fig. 4). Meanwhile, in FDO ORP$_{mean}$ began to increase significantly 1.5 hours after the DP had started (t = 25.5 h, Fig. 4), following the decrease in the VWC along the column (Fig. 2a, S1a, S2a) and the atmospheric air penetration. It is important to note that for roughly 1.33 hours after the pump was turned off, the soil was still covered with effluent, not allowing significant air penetration (Fig. S5).






Both the air pulses in AI-LF$_1$ and the DP in FDO led to a recovery of the ORP levels in the soil. However, this recovery was only partial. In FDO, the maximal ORP$_{mean}$ during the first FP was 306 mV, while during the second FP maximal ORP$_{mean}$ deteriorated to 179 mV. While the passive aeration in this experiment successfully recovered the ORP near the surface (depths of 25 and 65 cm below soil surface; Fig. S4a, s4b), in the deepest measured depth (145 cm below soil surface) the ORP remained low during the DP (Fig. S4d). Turkeltaub et al. (2022) found a similar trend at the SHAFDAN SAT site in Israel, where at one of the measurement stations, neither short nor longer DPs (~2, ~3 days, respectively) did not manage to recover the ORP at 100 cm depth.

In contrast, the active aeration in AI-LF$_1$ succeeded where the passive aeration failed - the ORP at 145 cm depth increased immediately in response to the air pulses (Fig. S4d), even though the injection was performed 60 cm above (Fig. 1). This observation supports our pre-described hypothesis that direct air movement downwards occurred simultaneously to the major air movement upwards. However, in AI-LF$_1$, ORP$_{mean}$ generally declined with time. The maximal ORP$_{mean}$ during the first 24 hours was 357 mV, and following the first and the second pulse maximal ORP$_{mean}$ was 277 and 266 mV, respectively (Fig. 4). This overall deterioration indicates that this operation mode is unsatisfactory for reliable maintenance of aerobic WW treatment. Extending the pulse duration may lead to better recovery, but the tradeoff is lower infiltrated volumes (as there is a longer period characterized by diminished infiltration flux) and higher energy costs.

AI-HF$_1$ included much shorter pulses (8 min) than AI-LF$_1$ (60 min; Table 1). However, the pulses' frequency (every 4.5 hours) did not allow ORP$_{mean}$ to deteriorate to the range of the highly negative values, which were observed during FDO and AI-LF$_1$. The minimal ORP$_{mean}$ observed during AI-HF$_1$ was -23 mV, while in FDO and AI-LF$_1$, it was -119 and -192 mV, respectively (Fig. 4). In AI-HF$_1$, the deterioration in the ORP levels with time was much less drastic than in AI-LF$_1$. In fact, during the last 48 hours, there was even an overall trend of increase in ORP$_{mean}$. In summary, the high-frequency operation mode (AI-HF$_1$) maintained higher and steadier ORP levels compared to the low-frequency operation (AI-LF$_1$) and the intermittent operation (FDO; Fig. 4).





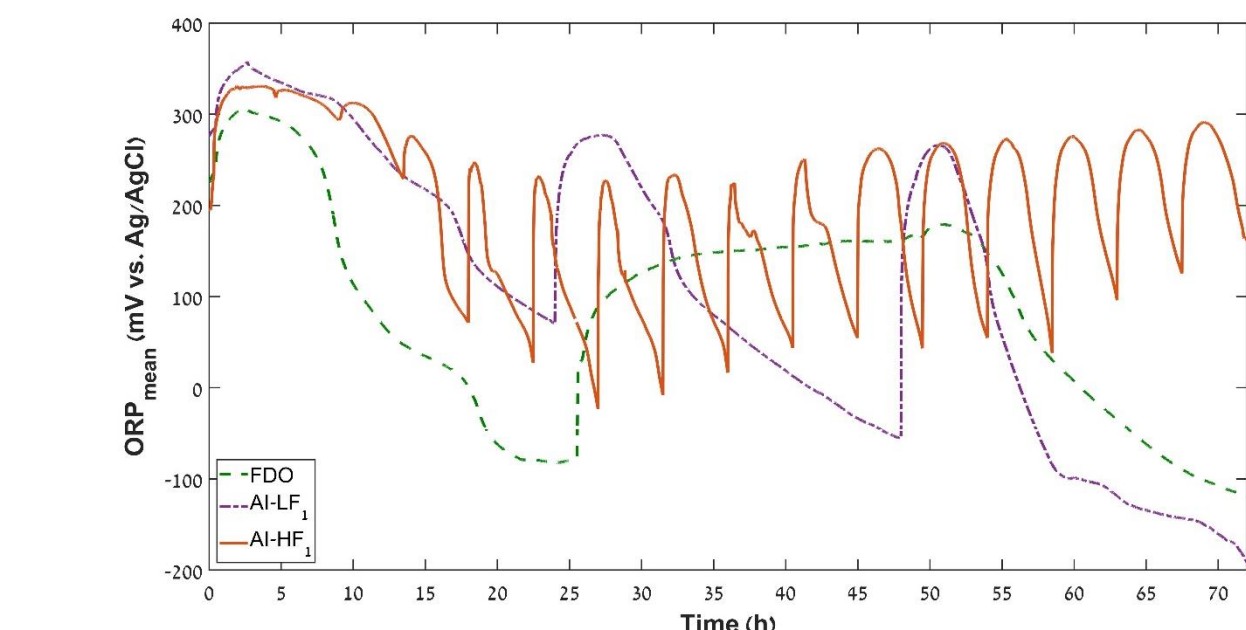

**Fig. 4.** $ORP_{mean}$ (arithmetic mean of the ORP measured at depths of 25, 65, 105 and 145 cm below soil surface) during FDO, AI-LF$_1$ and AI-HF$_1$.

Generally, the better and steadier oxidizing conditions in AI-HF$_1$ led to better and steadier effluent quality compared to AI-LF$_1$ and FDO (Fig. 5). For example, the normalized DOC concentration at a depth of 145 cm ($DOC_{145}$ / $DOC_0$) was lower in AI-HF$_1$ (mean = 0.05; i.e., 95% removal) than in FDO (mean = 0.14; Fig. 5a). A Mann-Whitney U (MWU) test indicated that this difference is statistically significant (p = 0.04, α = 0.05). As discussed earlier, DOC degradation in SAT is enhanced under oxidizing conditions; therefore, it should be expected that the highest DOC removal would be observed in the most oxidizing experiment (AI-HF$_1$). Correspondingly, the low-frequency operation mode (AI-LF$_1$), which showed the most negative ORP levels (Fig. 4), also showed the highest DOC concentrations at a depth of 145 cm below ground level (mean = 0.23). As discussed earlier, the formation of PFP and the reduced detention time in AI-LF$_1$ might also harm the obtained effluent quality in this experiment.

The normalized $DOC_{145}$ in AI-HF$_1$ was not only lower but also steadier (SD = 0.01) than in FDO and AI-LF$_1$ (SD = 0.08 and 0.18, respectively; Fig. 5a). Unsurprisingly, the worst (i.e., highest) $DOC_{145}$ values were observed when there was no DO throughout the column (i.e., $DO_{mean} = 0$), and $ORP_{mean}$ was lower than 75 mV (indicated by filled dots in Fig.5). When at least some parts of the soil were characterized by aerobic conditions (i.e., $DO_{mean} > 0$ and $ORP_{mean} > 90$ mV, indicated by unfilled dots), lower concentrations were observed (Fig. 5a).



The preferred oxidizing conditions in AI-HF$_1$ also promoted the removal of reduced forms of N (TKN): the normalized TKN$_{145}$ was lower in AI-HF$_1$ (mean = 0.11) than in the conventional FDO (mean = 0.51; Fig. 5b). We expected these results since as

discussed earlier, TKN is biologically removed in SAT through mineralization of organic-N to NH$_4^+$ and the following nitrification, which are both enhanced under aerobic conditions. However, it should be noted that the difference in the normalized TKN$_{145}$ between AI-HF$_1$ and FDO was statistically less significant than the difference in the DOC$_{145}$ (MWU test, p = 0.08, α = 0.05). Like the DOC, the TKN concentrations in AI-HF$_1$ were relatively steady throughout the experiment, while in FDO and AI-LF$_1$, low ORP levels and O$_2$ absence in parts of each experiment (Fig. 4, S6, respectively) led to poor TKN

removal (Fig. 5b). In terms of TKN, AI-LF$_1$ again showed the poorest effluent quality – less than 40% of removal on average (Fig. 5b).

The main product of the coupled mineralization-nitrification is NO$_3^-$. Hence, it can be expected that high NO$_3^-$ concentrations will be observed when these processes are enhanced and lead to high TKN removal. Indeed, in AI-HF$_1$, the mean NO$_3^-{}_{(145)}$ was

~5.5 times higher than its mean inlet concentration (Fig. 5c). This finding is not unusual – an increase in NO$_3^-$ concentration in the first meters of SAT systems was also observed in previous lab experiments and in the field (Grinshpan et al., 2022; Sallwey et al., 2020; Wilson et al., 1995). Meanwhile, in FDO and AI-LF$_1$, the mean NO$_3^-{}_{(145)}$ concentrations were much lower (Fig. 5c), as the TKN removals were also lower (Fig. 5b). The difference in the normalized NO$_3^-{}_{(145)}$ between AI-HF$_1$ (mean = 5.63) and FDO (mean = 0.45) was statistically significant (MWU test, p = 0.0007, α = 0.05). As opposed to the DOC and

TKN, lack of O$_2$ led to low NO$_3^-{}_{(145)}$ concentrations (Fig. 5c). This makes sense since nitrification (NH$_4^+$ to NO$_3^-$) is an aerobic process that does not occur under O$_2$ absence, while these anoxic conditions promote NO$_3^-$ removal by denitrification (Mienis and Arye, 2018). In regions where nitrate levels are of major concern, this should be taken into account.

Since N removal by denitrification is an anoxic process, it could be expected that TN concentrations in the mostly aerobic

AI-HF$_1$ would be higher than in FDO and AI-LF$_1$. However, the normalized TN$_{145}$ during AI-HF$_1$ (mean = 0.58) was not statistically different (MWU test, p = 0.60, α = 0.05) from the normalized TN$_{145}$ during FDO (mean = 0.50) and even lower than the normalized TN$_{145}$ in AI-LF$_1$ (mean = 0.64; Fig. 5d). In addition, it should be noted that the worst N removals (i.e., highest TN concentrations) were observed when O$_2$ was absent from the soil (Fig. 5d). These findings indicate that although O$_2$ absence is vital for efficient denitrification, O$_2$ presence is essential for N removal in SAT, as aerobic nitrification is a

necessary stage in converting reduced forms of N (organic-N and NH$_4^+$) to N$_2$. In other words, reducing conditions in SAT restricts the creation of NO$_3^-$ and can turn it into a limiting factor of N removal by denitrification.

In this study, the air injection significantly influenced the O$_2$ availability and the ORP above and below the injection port, creating, in some cases, a fully oxidized soil profile (Fig. S4, S6) which inhibited N removal by denitrification. However, in

real SAT sites, where the soil profile is much deeper and larger, the impact of shallow air injection on deep parts of the soil is expected to be negligible, as due to buoyancy, the injected air will probably move mostly upward. Hence, in the field, air





injection into the subsurface may divide the vadose zone into two pseudo reactors – the upper one, located above the injection port, will be characterized by stable oxidizing conditions that enhance aerobic DOC degradation, organic-N mineralization and nitrification. The lower one, located below the injection port, will be characterized by anoxic conditions that enhance

denitrification. As a result, Air-SAT may lead to even better N removal than presented here. This, however, is yet to be tested.

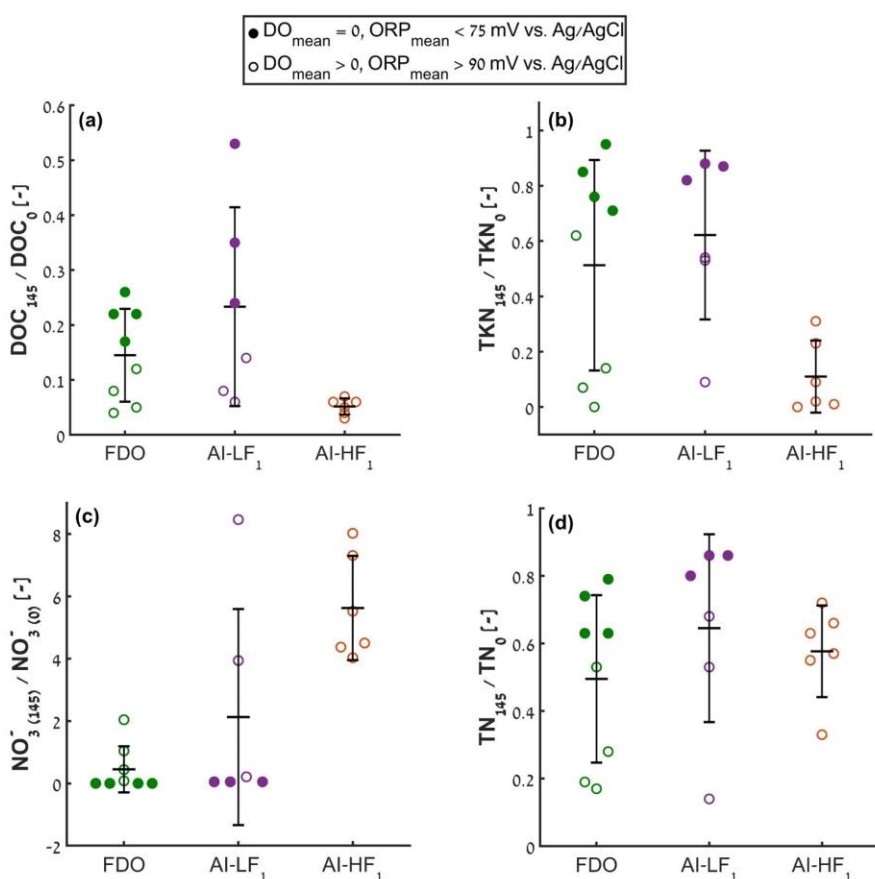

**Fig. 5.** Concentrations of DOC **(a)**, TKN **(b)**, $NO_3^-$, **(c)** and TN **(d)** at a depth of 145 cm during FDO, AI-LF$_1$ and AI-HF$_1$. All concentrations are normalized to the inlet concentration of the same species. The horizontal line represents the arithmetic mean and the error bars show plus and minus one SD. Values below quantitation limit (QL) were regarded with QL/2. QLs of the relevant species are available in the supplement (Table S7). Filled dots indicate samples taken when the mean DO throughout the column was zero and the mean ORP was lower than 75 mV, while unfilled dots indicate samples taken when the mean DO was higher than zero and the mean ORP was higher than 90 mV.

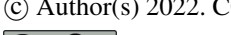



The high-frequency air injection also led to consistently higher removal of the emerging contaminant IBP (Fig. 6a): the normalized $IBP_{145}$ was lower in $AI-HF_1$ (mean = 0.12) than in FDO (mean = 0.77). This difference was statistically significant (MWU test, p = 0.01, α = 0.05). In addition, like the DOC and TKN, the IBP concentrations also showed dependency on the soil's ORP and $O_2$ presence (Fig. 6a). Numerous studies, such as Amy & Drewes (2007), He et al. (2020) and Onesios &

Bouwer (2012), found that IBP is efficiently removed during SAT. Sallwey et al. (2020) also demonstrated that its efficient removal (>80%) is preserved under different flooding-drying regimes, suggesting that IBP is relatively insensitive to DO fluctuations throughout the soil profile. However, our results indicate that although IBP can be removed efficiently (>80%) even under mean DO levels that are smaller than 1 mg $L^{-1}$ (Fig. S7), when $O_2$ is totally absent, its removal is poor (<40%; Fig. 6a, S7). Hence, to maintain a steady-high removal of IBP, it is necessary to preserve aerobic conditions in the top layers of

SAT sites, either by passive aeration (i.e., DPs) or active aeration.

The other two examined CECs showed much poorer removal than IBP. Furthermore, they showed weak dependency on the soil redox conditions and on the operation mode itself (Fig. 6b, 6c). CBZ mean removal was less than 10% during the three main experiments (Fig. 6b), while BTR was removed only by 1-21% on average (Fig. 6c). The CBZ removal we observed

matches with findings from previous studies, which showed that CBZ is very persistent during SAT (Usman et al., 2018; Sopilniak et al., 2018; He et al., 2020), while its limited removal is associated mainly with sorption rather than with biodegradation (Martínez-Hernández et al., 2016). Hence, the air injection, which enhanced aerobic biodegradation but probably did not have a meaningful impact on sorption processes, also did not improve the CBZ removal.

With respect to BTR, previous studies' results are equivocal and site-specific. Filter et al. (2017) and Wünsch et al. (2019) reported poor BTR removal (<30%) during column experiments with sediment cores from the Saatwinkel SAT site in Germany and soil from the Lange Erlen site in Switzerland, respectively. On the other hand, Glorian et al. (2018) reported 77-98% of BTR removal in bank filtration sites in Northern India. In addition, Rodríguez-Escales et al. (2017) and Sallwey et al. (2020) demonstrated that BTR removal in SAT depends on soil redox conditions, where aerobic conditions are preferable. Hence,

although the BTR removal was insufficient in our study and the air injection did not improve it, we believe it may be beneficial in different circumstances. For example, it is possible that in our study, the limiting factor of the BTR biodegradation was not DO availably, but other factors such as lack of proper microbial community or specific nutrients. In SAT sites where these are abundant, low DO concentrations may hinder BTR removal. In such conditions, active air injection can induce steady aerobic conditions that will enhance the biodegradability of BTR and lead to better effluent quality.


In contrast to recharge volumes (Fig. 3), the airflow rate did not significantly impact the obtained effluent quality in the outflow: the concentrations of the macro pollutants and the CECs at a depth of 145 cm in the secondary experiment $AI-HF_2$ were similar to $AI-HF_1$ (Table S6, S7). This experiment (i.e., $AI-HF_2$) was identical to $AI-HF_1$ in terms of hydraulic operation (continuous infiltration) and pulse frequency ($4.5^{-1}$ $h^{-1}$), but the average airflow rate was approximately half (0.991 SLPM). In





addition, the pulse duration in this experiment was double (16 min) than in AI-HF₁ (8 min) in order to achieve a similar overall

volume of injected air (Table 1), which allows a fair comparison in terms of effluent quality.


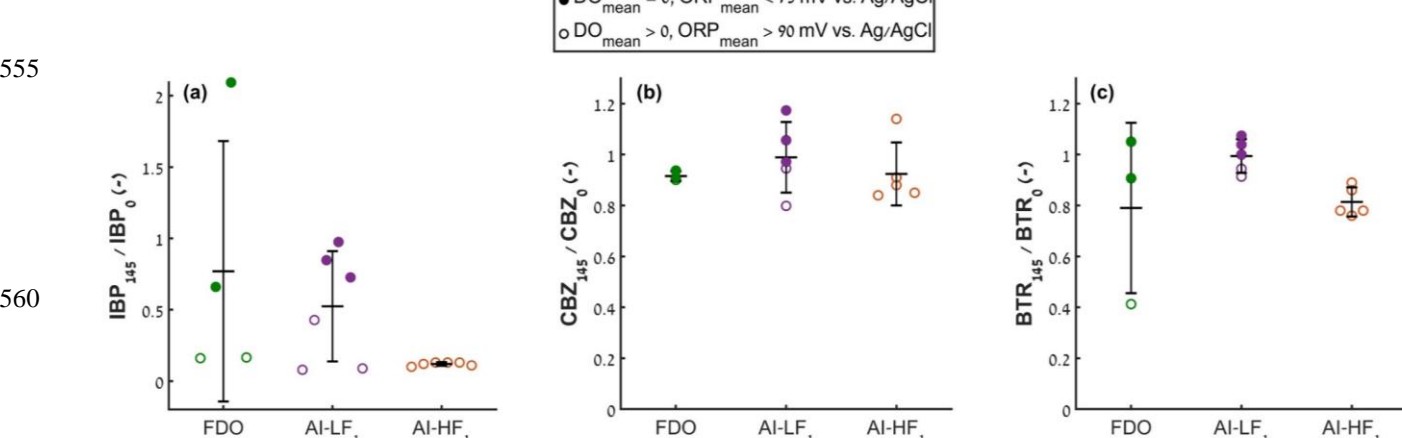


**Fig. 6.** Concentrations of IBP **(a)**, CBZ **(b)** and BTR **(c)** at a depth of 145 cm during FDO, AI-LF₁ and AI-HF₁. The concentrations are

normalized to the inlet concentration of the same species. The horizontal line represents the arithmetic mean and the error bars show

plus and minus one SD. Values below detection limit (DL) were regarded with DL/2. DLs of the relevant species are available in the

supplement (Table S7). Filled dots indicate samples taken when the mean DO throughout the column was zero and the mean ORP was

lower than 75 mV, while unfilled dots indicate samples taken when the mean DO was higher than zero and the mean ORP was higher

than 90 mV. In FDO, samples were taken and analyzed for only one replicate. In addition, for CBZ and BTR, the first sample from each

experiment (taken at t ~ 4.3 h) was excluded as an outlier since it showed much lower concentrations than the others, apparently due to

temporary retardation of the compound as a result of its interactions with soil components.

## 4 Summary and conclusions


The use of subsurface air injection as an alternative to long DPs operation in SAT (Air-SAT) was tested in a series of long-

column experiments. Synthetic effluent was introduced into the column for 72 hours continuously. At the same time,

compressed air was injected in pulses, at different durations (8, 16 and 60 min), frequencies (4.5⁻¹ and 24⁻¹ h⁻¹) and flow rates

(~1 and ~2 SLPM), at a depth of 85 cm below the soil surface. ORP, VWC and DO were monitored along the column, while

effluent samples were collected at various depths and analyzed for N species, DOC and selected CECs (IBP, CBZ and BTR).

In addition, the injection pressure, temperature, infiltration rates and overall infiltrated volumes were measured throughout

each experiment. All the results were compared to a conventional flooding-drying operation.

The Air-SAT operation allowed more effluent infiltration than the flooding-drying regime. The significantly increased

recharge volumes were achieved mainly due to the double infiltration time. In addition, the infiltration rate, which was



decreased by ~50% during the air pulses, recovered after them, and in some cases, even reached significantly higher values than in the conventional intermittent operation, probably due to the creation of new PFP for both effluent and air. Moreover, we found that longer pulse duration and higher airflow rate led to increased infiltration rate. In contrast, the high pulse frequency led to a lower infiltration rate than observed in the low-frequency operation, apparently owing to enhanced biological
clogging.

Similarly to the DPs, the air injection managed to recover the ORP and DO levels along the soil profile. In fact, high-frequency Air-SAT during continuous flooding maintained better and steadier oxidizing conditions than the flooding-drying operation. These conditions led to higher removal of DOC, TKN and IBP, similar removal of total N, but higher concentrations of $NO_3$.
On the other hand, low-frequency Air-SAT, which excelled in terms of infiltration rate, induced unsteady oxidizing conditions which led to similar or even worse effluent quality than the conventional operation.

In terms of effluent quality, short-pulses high-frequency Air-SAT seems to be the best operation mode. However, this operation mode induces steady aerobic conditions, which apparently, leads to enhanced biological clogging and infiltration rate
deterioration. In a long-term operation, this deterioration can be critical, and the DPs, which are known to fulfill a significant role in maintaining a steady infiltration rate in SAT, may reduce it. Hence, we suggest that the air injection may be operated during the conventional intermittent operation, allowing shorter DPs and higher recharge volumes while sustaining a steady infiltration rate.

This preliminary study opens the possibility to use subsurface soil air injection as an alternative to increase the recharge capacity of existing SAT sites while maintaining high effluent quality. Subsurface air injection may also solve some of the potential problems of Ag-SAT where appropriate DO and ORP conditions are important for crop health (Grinshpan et al., 2022, 2021). However, to reach a point of full application, further research is necessary, including a pilot-scale study, a techno-economic assessment and optimization of the operational parameters (i.e., injection ports' spread across the basin, injection
depth, airflow rate, pulse frequency and pulse duration).

*Data availability*. The data that support the findings of this study are available at https://doi.org/10.5281/zenodo.7265560 or upon request from the corresponding author.

*Author contributions*. IA, AZ and SBM prepared the experimental setup. IA, SBM, AF and NW designed the experiments, while IA and AZ carried them out. IA analyzed the data and prepared the paper with the contribution of all authors.

*Competing interests*. The authors declare that they have no conflict of interest.



*Acknowledgments.* The authors wish to thank Dr. Nura Azzam for the analytical assistance.

*Financial support.* This work was financed within the framework of the German-Israeli Water Technology Cooperation Program under the project numbers WT1601/2689, by the German Federal Ministry of Education and Research (BMBF) and the Israeli Ministry of Science, Technology and Space (MOST). This work was also supported by the Israel-U.S. Collaborative
Water-Energy Research Center (CoWERC) via the Binational Industrial Research and Development Foundation (BIRD) Energy Center, grant EC-15.

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
