# Peer review of "Improving soil aquifer treatment efficiency using air injection into the subsurface"

_Hydrology and Earth System Sciences, 2022_

## Author Comment (AC1)

**Referee #1**

We thank the referee for his supportive review and for the useful comments. Below are our answers, including proposed action, *following each of the comments.*

1. *Line 18, abstract: Please add the length of the columns in the column description (e.g. after six long column...)*

    Column length was 2 m, as described in the text. This information was added to the abstract (P.1. line 18).

2. *Line 24: "doubled the infiltration time" might be misleading as wording. Suggest changing to "double the time during which infiltration was possible"*

    Accepted and was rephrased in the revised manuscript (P.1, line 25).

3. *Line 27: What is SLPM?*

    SLPM stands for *standard liter per minute* (i.e., at given temperature and pressure). We changed the wording and now use l/min instead and clarify this in the revised manuscript (P. 1, line 28 and throughout the manuscript).

4. *Lines 59-62: Could you please state threshold values or value ranges for oxygen for some of the processes listed here? What do you consider aerobic conditions? Under what oxygen conditions does the ammonification rate decrease substantially? Is there a threshold that can be defined as clear tipping point? Does nitrification assume the same oxygen conditions (same number ranges)?*

    This is an interesting and important question. While in this study we did not use precise threshold values, the literature contains several different definitions (e.g., Sohasalam et al., 2008 who used the ranges of above 300 mV, 100-300 mV, and below 100 mV, respectively). Other classifications (e.g., Zhang and Furman, 2021 who followed several other definitions) classified Redox differently (oxidizing, weakly reducing, moderately reducing, and strongly reducing conditions (>400, 200 to 400, -100 to 200 and <-100 mV, respectively). The following text was added to the revised manuscript (P. 3, line 69-70). *"[Note, we follow here the definitions of Sohasalam et al., 2008 who used the ranges of above 300 mV, 100-300 mV, and below 100 mV, respectively]."*

5. *Line 72: What is the mechanisms behind the higher removal of CECs during longer drying periods? Is it just the aerobic conditions?*

    We are not sure if the reviewer refers here to CECs or to removal of major carbon and nitrogen compounds. The specific mechanism of CEC degradation was not part of this

research but is discussed in detail starting at P.17, line 506. The literature suggests (as discussed in the later parts of the results and discussion section) that in some cases it is related to adsorption and in some to oxidation. Our results support the oxidation approach. However, it would be premature to suggest so with such a minimal database.

6. *Line 82: The column description is a bit confusing. Was the column only 200 cm long with the top 30 and the bottom 30 cm kept empty to provide room for flooding and drainage. That would suggest only 140 cm were effectively filled with soil? Also what are the 20x10x10 modules? How were they connected? How did you pack the column and avoided capillary boundaries between the layers/modules?*

   Thanks for highlighting the lack of clarity. The column itself is 200 cm long, of which 160 cm was filled with soil. The upper 30 cm was operational, and the lower 10 cm was used for drainage. This was clarified in the revised manuscript (P.4, lines 92-93).

   The column is modular with perfect sealing between modules (both rubber ring, shape fit, and overall pressing of the column with external long screws from top to bottom). This information was also added at P.4 lines 94-95.

7. *Line 108: By "surface pressure head" you mean the hydrostatic pressure of the ponded water?*

   Exactly. This is now clarified in the revised manuscript (P.5, line 114).

8. *Line 158: SLPM still not defined …but finally given in Table caption for Table 1.*

   See answer to comment #3. We have changed to l/min throughout the manuscript.

9. *Line 196: Why calculate a mean ORP for the profile if there is clearly a gradient with depth?*

   Mean ORP is calculated only for the purpose of the discussion regarding overall system performance. In other places we use specific ORP values. This was explained in the revised manuscript: "*To clarify, mean ORP is calculated only for the purpose of discussion of overall system functionality*" (P.7, lines 189-190).

10. *Line 199: Why did you choose to use glucose, which is a very digestible form of carbon? What forms of carbon are typically found in the wastewater?*

    The reviewer is touching here one of the greatest technical challenges of this research. First, wastewater contains a cocktail of various forms of carbon molecules, as noted in the manuscript with reference for a specific relevant source. We have attempted first to use several "well-accepted wastewater recipes" that contain other forms of carbon (see below, experiment conducted by Dr. Zengyu Zhang, in our lab), and found that they, in many cases, degrade too fast (even before reaching the soil). These simple experiments show that

glucose is reasonable choice in terms of degradation time. We agree that this is not ideal, but as can be seen in the figure, reasonable. To clarify that point, we have added (P.7, lines 194-195) the following text: "*Although not ideal, the use of glucose is common in wastewater degradation studies (e.g., Liu & Logan, 2004; Liao et al., 2001)*". Our plan for the coming year is to work with treated wastewater from a treatment plan at larger setup. This is impossible in laboratory experiments for various practical and technical reasons.

[Figure]

11. *Line 283: The fast developing clogging effect is really striking! Have you thought about adding soil microbial analysis to investigate whether the abundance of bacteria is increasing supporting the idea of biofilm formation?*

    Thanks for the suggestion. During the experiments described in this manuscript it was not possible to take soil samples. Nevertheless, future experiments will include vast microbiology analyses and quantification, including the correlation between the development of biofilm and clogging.

12. *Line 385: These results suggest that the air perhaps should be injected even deeper (>1m) in the profile? Air buoyancy will ensure that air rises to the top of the profile but perhaps the deeper injection can address some of the low oxygen concentrations in the deeper profile.*

    This is true, but not 100% precise. Deeper injection also means that the air pressure will have to be higher than the hydrostatic pressure at that depth, which may require techno-economic optimization. Moreover, practically, inserting air tubes in depth of more than 1 m will be expensive and more complicated technically (i.e., different types of machinery). At the same time, greater depths probably mean larger space between air-injection tubes, which may lower the costs. These points will be examined in our future field-scale research.

13. *Line 465: It seems maintaining aerobic conditions in the upper profile provides suitable conditions for mineralization and nitrification and having low oxygen and reduced conditions in the deeper profile creates the right environment for denitrification. Often natural soils*

*show a huge decrease in microbial abundance from the top 10 cm of the soil profile to 1 m.*

*What is the microbial abundance on denitrifiers in the deeper soil profile that can actually*

*reduce large amounts of nitrate?*

Thanks for this interesting question. This specific question was not tested in our

experiments, and therefore we cannot answer directly. As mentioned in our answer to

comment 11, we do intend to test in detail the microbiological aspects in a series of field

experiments that we hope to operate in the coming year.

Sohsalam, P., & Sirianuntapiboon, S. (2008). Feasibility of using constructed wetland treatment for molasses wastewater treatment. Bioresource Technology, 99(13), 5610-5616.

Zhang, Z., & Furman, A. (2021). Redox dynamics at a dynamic capillary fringe for nitrogen cycling in a sandy column. *Journal of Hydrology*, *603*, 126899.

Liao, B. Q., Allen, D. G., Droppo, I. G., Leppard, G. G., & Liss, S. N. (2001). Surface properties of sludge and their role in bioflocculation and settleability. Water research, 35(2), 339-350.

Liu, H., & Logan, B. E. (2004). Electricity generation using an air-cathode single chamber microbial fuel cell in the presence and absence of a proton exchange membrane. Environmental science & technology, 38(14), 4040-4046.

---

## Author Comment (AC2)

**Referee #2**

We thank the referee for his supportive review and for the useful comments. Below are our answers, including proposed action, *following each of the comments.* This includes the general comment regarding the treatment of interpretation as interpretations and not as results (comment #9).

1. The manuscript uses many abbreviations. This is a question of taste, of course, but I had to scroll back and forth a lot during reading. A table might be useful.

   Thanks for this suggestion. A list of abbreviations was added after the abstract (P.2, line 39)

2. *Figure 1 and Section 2.2: It did not become clear to me how the infiltration was carried out. I might have missed the description. From the text I understood that the infiltration was done by a constant head that was realized with the overflow tank. The resulting infiltration rates were not constant and depended on the soil conditions. In Figure 1, there is a peristaltic pump that seems to inject water with a constant rate. If water was pumped with constant rate, I would not understand how this could be, as the infiltration rate was measured and was not constant.*

   The referee probably missed the overflow pipe (w3 in Fig. 1).

3. *Line 145-146: I did not understand the sentence. What is meant by no pressure head above the unsaturated soil surface? Does this mean no ponding water? If there is water, the water always has a pressure head, which can be zero or positive or negative, but there cannot be no pressure head. I guess what is meant is no water.*

   Thanks for pointing out this clarity issue. We have rephrased the sentence and it now reads "*Six 72-hour column experiments were conducted. All experiments started with a dry system (i.e., after at least nine days of drainage), and no ponding on top.*" (P.5, lines 136-137).

4. *Line 146: I also did not understand how the effluent supply can be higher than the infiltration. From figure 1 I understand that the effluent is the infiltrated water. Or were there separate supplies of effluent and clear water?*

   The water (effluent) supply is constant, and almost always at a rate higher than the soil's capacity. This leads initially to ponding, and then to overflow, as described in the text (P.5 lines 137-138)

5. *Line 191: The 'h' is missing after the times.*

   Thanks. Corrected (P.7, line 182)

6. *Lines 192-194 and Figure 2: I was here also confused about the infiltration rates. If I understand correctly, the infiltration rates were concluded from the volume in the tank. How often was this done? In Figure 2, the mean infiltration flux is given and I assume that this is the time average of the infiltration flux. At least this would explain why it is so constant over longer time spans in figure 2. What time intervals were chosen for averaging? And why? Why not showing the time series of the infiltration rate without averaging? This would allow to get a better picture of how much the infiltration rate is related to the water content in the upper soil column part.*

Our intention was to use a pressure transducer to measure infiltration in high temporal resolution, but due to technical difficulties it did not function during some of the experiments, and we chose not to present different data for different experiments. The average is for an entire period, which is 24 h for FDO, 23 or 24 and 1 h for AI-LF1, and 24 h for AI-HF1. This is now clarified in the text: "*As a pressure transducer was not functioning for some of the experiments, the fluxes are averaged over time as presented in Fig. 2.*" (P.7, lines 185-186)

7. *Related to the previous comment: If I understood the averaging of the infiltration correctly, I think it would be better to use the term 'mean infiltration' consistently in the text instead of calling it 'infiltration' (example lines 275, 279). This would prevent misunderstanding.*

Thanks. Indeed, using mean infiltration is more appropriate and we have changed that at 20 different locations throughout the text.

8. *Section 3: Do I understand it right that the first 24 hours of the experiments FDO and AI-LF1 were the same? The large difference of water content in the upper soil at the beginning is thus the inherent uncertainty in the experimental results. I think this has to be considered when conditions in the experiments are compared. Smaller deviations have to be interpreted a bit carefully.*

The observation is correct – FDO and AI-LF1 are identical in terms of operation in the first 24 h, other than small changes in the initial conditions. However, we disagree that the differences are high. For most of the time the water content in both experiments is 28%, with FDO increasing to 35% at about 18 h, with practically no difference in recharge throughout the period. The main difference is probably the earlier release of entrapped air in FDO. This is clearly explained in the text (P.8-9, lines 233-237).

9. *Lines 259-266: This paragraph contains an interpretation of the increasing water content and infiltration rates with time in the air injection experiments with low infiltration rates. The interpretations are plausible, however, they are just interpretations. There are other*

*possibilities that could explain the findings, such as redistribution of water with the air injection that leads to connected patterns with higher water saturation and thus higher conductivity, facilitating higher infiltration. The increased water content could be local, and would then be rather accidentally. Also, the interpretation with the breaking of crusts reads a bit odd. I am not an expert on bio-clogging, but the time seems very fast to me. If breaking of clogging crusts was relevant for the increase of injection, the crusts must have formed before the first 24 hours. Is it realistic that this happens so fast? Is there evidence for crust building in the sand column (maybe visible at the surface)? I am also not so convinced by the preferential pathways. Why should a re-arrangement of grains (this is how I would interpret the 'creating wider pores' read) should lead to large pore clusters connected from top to bottom? It is possible, but not very likely. To clarify: The interpretations are plausible and I would not argue against them. But the preferential pathways and crusts are later in the manuscript treated as results. I think they need to be kept as possible processes that take place, not as given ones.*

Thanks for this observation. Our empirical observations indicate that in SAT systems crusts do form (biological, but also physical due to compaction under very high hydraulic loads), but clearly this phenomenon was not specifically looked at in our experiment. To clarify that, we have added the following sentence:" *It is not certain that crusts were formed in this controlled experiment, but they do exist in SAT systems*." (P. 9, line 254). We have also changed the text accordingly in two other locations (P.10, line 279; P11, line 320).

10. *Line 278: If the water content in the whole column decreased, where did the water go? Could one observe a larger outflow from the column that matches the decreased water content?*

Thanks for this observation. Drainage from the system was not measured in high enough temporal resolution to confirm enhanced drainage (which makes sense, considering the elevated air pressure), but clearly during air injection water drained from the system bottom, while the air did not allow further water recharge from top to compensate for that drainage. The following text (P.10, line 271-272) was added for clarification: "*The drop in water content is due to air impeding infiltration, while gravitational drainage continues (drainage measurements were not conducted in high enough resolution to confirm enhanced drainage due to air pressure).*"

11. *Line 350-351: This conclusion is here a bit misplaced and should better be moved to the end, when also the water quality was discussed.*

Thanks for this suggestion. We have moved this conclusion (that also includes operational suggestion) from section 3.1 to the conclusions (P.19 lines 575-582)

12. *3.2: 'Effluent quality', not 'effluent quality'*

Typo corrected. Thanks (P.12, line 354)

13. *A general comment: Would it for practical application not be important to study the range of influence of an air injection point? In the experiments, air influenced the water quality in the whole column, but as is later written in the discussion, will probably have an influence only to a certain depth. The column had a not too big cross section, so that the air flow was directed mainly vertically. In a real field a large area would have to be reached. This might involve a lot of injection points.*

This is very accurate. Our intention in a follow-up experiment, in field conditions and with non-synthetic wastewater, is to experiment with both depth and separation of air injection laterals. Such data, used to calibrate a numerical model, would allow upscaling and optimization at full scale. This intention is clearly mentioned in the conclusions (P. 19, line 600-602)